# Deletion of Osteopontin Enhances β_2_-Adrenergic Receptor-Dependent Anti-Fibrotic Signaling in Cardiomyocytes

**DOI:** 10.3390/ijms20061396

**Published:** 2019-03-20

**Authors:** Celina M. Pollard, Victoria L. Desimine, Shelby L. Wertz, Arianna Perez, Barbara M. Parker, Jennifer Maning, Katie A. McCrink, Lina A. Shehadeh, Anastasios Lymperopoulos

**Affiliations:** 1Laboratory for the Study of Neurohormonal Control of the Circulation, Department of Pharmaceutical Sciences (Pharmacology), College of Pharmacy; Nova Southeastern University, Fort Lauderdale, FL 33328, USA; cp1743@mynsu.nova.edu (C.M.P.); vd359@mynsu.nova.edu (V.L.D.); sw1541@mynsu.nova.edu (S.L.W.); ap2491@mynsu.nova.edu (A.P.); barbaramparker@gmail.com (B.M.P.); jm3706@mynsu.nova.edu (J.M.); km1911@mynsu.nova.edu (K.A.M.); 2Department of Medicine, University of Miami Miller School of Medicine, Miami, FL 33136, USA; LShehadeh@med.miami.edu

**Keywords:** β_2_-adrenergic receptor, cAMP, cardiac myocytes, CRISPR, Epac1, fibrosis, osteopontin, signal transduction

## Abstract

Cardiac β_2_-adrenergic receptors (ARs) are known to inhibit collagen production and fibrosis in cardiac fibroblasts and myocytes. The β_2_AR is a Gs protein-coupled receptor (GPCR) and, upon its activation, stimulates the generation of cyclic 3′,5′-adenosine monophosphate (cAMP). cAMP has two effectors: protein kinase A (PKA) and the exchange protein directly activated by cAMP (Epac). Epac1 has been shown to inhibit cardiac fibroblast activation and fibrosis. Osteopontin (OPN) is a ubiquitous pro-inflammatory cytokine, which also mediates fibrosis in several tissues, including the heart. OPN underlies several cardiovascular pathologies, including atherosclerosis and cardiac adverse remodeling. We found that the cardiotoxic hormone aldosterone transcriptionally upregulates OPN in H9c2 rat cardiac myoblasts—an effect prevented by endogenous β_2_AR activation. Additionally, CRISPR-mediated OPN deletion enhanced cAMP generation in response to both β_1_AR and β_2_AR activation in H9c2 cardiomyocytes, leading to the upregulation of Epac1 protein levels. These effects rendered β_2_AR stimulation capable of completely abrogating transforming growth factor (TGF)-β-dependent fibrosis in OPN-lacking H9c2 cardiomyocytes. Finally, OPN interacted constitutively with G_α_s subunits in H9c2 cardiac cells. Thus, we uncovered a direct inhibitory role of OPN in cardiac β_2_AR anti-fibrotic signaling via cAMP/Epac1. OPN blockade could be of value in the treatment and/or prevention of cardiac fibrosis.

## 1. Introduction

Cardiac fibrosis is a fundamental process mediating the adverse remodeling of the heart post-myocardial infarction (MI) or other ischemic injury, as well as in heart failure (HF) [1]. The main cell type driving the fibrotic process is activated myofibroblasts, with significant contributions by monocytes/macrophages, lymphocytes, mast cells, vascular endothelial cells, and cardiomyocytes. All of these produce and secrete key pro-fibrotic factors such as reactive oxygen species, proteases, and metalloproteinases; and fibrosis-promoting growth factors, predominantly transforming growth factor (TGF)-β [2]. 

β-adrenergic receptors (ARs) are the main receptors mediating most actions of the catecholamine hormones norepinephrine and epinephrine in the heart [3,4,5]. All three βAR subtypes are expressed in the adult mammalian myocardium, with the β_1_AR being the most abundant and mediating the positive inotropic and chronotropic actions of the sympathetic nervous system, whereas the less-abundant β_2_AR exerts several cardio-protective effects in the post-MI heart, such as inhibition of apoptosis, inflammation, fibrosis, etc. [4,6]. Indeed, β_2_AR inhibits collagen production and fibrosis in cardiac fibroblasts [7,8,9]. All ARs belong to the G protein-coupled receptor (GPCR) superfamily and both β_1_AR and β_2_AR activate the Gs protein signaling pathway in cardiac cells, leading to the activation of adenylyl cyclase (AC) and subsequent cyclic 3′,5′-adenosine monophosphate (cAMP) synthesis [3]. cAMP is a major second messenger inside cells, and in cardiac myocytes it is responsible for the stimulation of cardiac contractility thanks mainly to activation of its effector kinase PKA (protein kinase A or cAMP-dependent protein kinase), which phosphorylates a variety of substrates to facilitate contraction in the cardiac myocyte [4,10,11]. In addition to PKA, cAMP also directly activates Epac (exchange protein directly activated by cAMP) [12,13]. Two Epac isoforms have been characterized: Epac1 and Epac2. Epac1 is expressed ubiquitously and is quite abundant in the heart, including in cardiac fibroblasts [14]. This Epac isoform has been documented to decrease collagen expression in response to βAR activation in rat cardiac fibroblasts and its expression is downregulated in the heart post-MI and in response to pro-fibrotic stimuli [12,14,15,16]. Epac1 blocks collagen and DNA synthesis in rat cardiac fibroblasts, and was very recently reported to also block atrial fibroblast activation, migration, and secretion of fibrotic mediators in post-MI mice and in HF dogs [9,14].

Osteopontin (OPN) is a member of the small integrin-binding ligand N-linked glycoprotein (SIBLING) family, and is expressed in normal mineralized tissues and cell types including osteoblasts, macrophages, lymphocytes, vascular smooth muscle cells (VSMCs), renal and cardiac fibroblasts, and several neoplastic tissues [17,18,19]. As a pro-inflammatory protein, OPN induces VSMC inflammation in the brain, pancreas, kidney, and heart [20]. It also participates in cell adhesion and migration processes via interaction with various integrins and CD44, and is a potent chemokine for mononuclear cells and VSMCs [20]. Thus, it comes as no surprise that OPN has been implicated in various pathologic conditions, including atherosclerosis [18,21,22,23], wound healing [19], and cardiac adverse remodeling [24]. In the present study, we sought to investigate OPN’s role in βAR-regulated cardiomyoblast fibrosis. Using the H9c2 rat cardiac myoblast cell line as our cell model, which endogenously expresses both β_1_AR and β_2_AR, as well as OPN [25,26], we found that OPN substantially impedes β_2_AR-induced, cAMP/Epac1-dependent anti-fibrotic signaling in cardiac cells. 

## 2. Results

### 2.1. Regulation of OPN Expression by Aldosterone and the β_2_AR in H9c2 Cardiomyocytes

Since aldosterone is a well-documented cardio-toxic hormone mediating adverse remodeling, including cardiac fibrosis [27], we first examined the effect of this mineralocorticoid on OPN expression in H9c2 cardiac cells. As shown in Figure 1, aldosterone treatment for 2 h caused a rapid upregulation of OPN mRNA levels, indicating that OPN is an immediate/early response gene for aldosterone in H9c2 cardiac cells, consistent with previous studies demonstrating this in renal and other cell/tissue types [20,28].

Additionally, this effect of aldosterone on OPN upregulation was mineralocorticoid receptor (MR)-dependent (data not shown) (i.e., a classic genomic/transcriptional effect of aldosterone in H9c2 cardiomyocytes). Interestingly, β_2_AR activation with salbutamol (albuterol), applied together with aldosterone, completely prevented the OPN mRNA induction by the latter (Figure 1). This finding strongly suggests that selective β_2_AR stimulation may inhibit (at least some of) the genomic, MR-dependent effects of aldosterone in H9c2 cardiac cells.

### 2.2. OPN Opposes β2AR cAMP Signaling in H9c2 Cardiomyocytes

Since the β_2_AR inhibits OPN upregulation in H9c2 cardiomyocytes (an effect consistent with this receptor’s anti-fibrotic actions in the heart), we posited that there is perhaps a negative feedback loop operating in cardiac myocytes allowing OPN to reciprocally regulate (inhibit) β_2_AR function and signaling. To this end, we deleted the OPN gene in H9c2 cells via CRISPR and compared the extent of cAMP accumulation—the major second messenger generated by βARs regulating cellular fibrosis [16]—between cells having OPN deleted (OPN-knockout (KO) cells) and wild type (WT) cells. After confirming the genetic deletion of OPN at the protein level (Figure 2A), we stimulated the cells with either isoproterenol to activate both β_1_ARs and β_2_ARs or salbutamol to selectively activate only the β_2_ARs and measured acute cAMP generation.

OPN KO led to a significant enhancement of cAMP synthesis in response to both isoproterenol and salbutamol stimulations in H9c2 cardiomyocytes, compared to WT myocytes (Figure 2B). Importantly, this was not due to differences in AC activity between WT and OPN KO cells, since forskolin—a direct AC activator—produced comparable levels of cAMP accumulation in both H9c2 cell clones (Figure 2C). Taken together, these results suggest that OPN opposes βAR-stimulated cAMP signaling—including β_2_AR-induced cAMP production—in H9c2 cardiomyocytes.

### 2.3. Epac1 Upregulation by OPN CRISPR-Mediated Deletion in H9c2 Cardiomyocytes

Since Epac1 plays an inhibitory role in cell/tissue fibrosis, including cardiac fibrosis [9,16], next we examined the effects of OPN KO on Epac1 protein expression. Treatment of WT and OPN KO H9c2 cardiac cells with either isoproterenol (or salbutamol, data not shown) for 24 h led to a significant upregulation of Epac1 protein levels in the absence of OPN, compared to WT cells (Figure 3A,B). This finding suggests that OPN not only blocks βAR-dependent cAMP production but also impedes βAR-induced Epac1 activation and upregulated expression in cardiac myocytes.

### 2.4. Prevention of Pro-Fibrotic Gene Expression by OPN Genetic Ablation in H9c2 Cardiomyocytes

To examine the impact of OPN’s β_2_AR signaling effects on the fibrosis of H9c2 cardiomyocytes, we measured the β_2_AR-dependent inhibition of collagen types I, III, and IV, as well as of fibronectin, on mRNA induction in response to the classic pro-fibrotic stimulus TGF-β1 in the presence (WT) and absence (OPN KO) of OPN in H9c2 cells.

Although, as expected, β_2_AR activation partially inhibited the transcriptional upregulation of TGFβ-induced collagen type I (Figure 4A), type III (Figure 4B), type IV (Figure 4C), and fibronectin (Figure 4D) in normal WT cells, the absence of OPN enabled salbutamol to completely abrogate the TGFβ-dependent expression of all of these pro-fibrotic factors. Thus, OPN significantly hindered the ability of the β_2_AR to block fibrosis via cAMP and Epac1 in cardiac myocytes. Of note, the partial inhibition of TGFβ-induced pro-fibrotic mediators by the β_2_AR in WT cells might be partly due to TGFβ-induced transcriptional upregulation of OPN itself, which could raise the “baseline” mRNA levels of collagen and fibronectin the β_2_AR would need to suppress in these cells.

### 2.5. OPN Inhibits β_2_AR cAMP Signaling by Directly Interacting with the G_α_s/olf Protein Subunit in H9c2 Cardiomyocytes

In an attempt to begin to dissect the molecular mechanism by which OPN perturbs β_2_AR’s cAMP-mediated anti-fibrotic signaling in H9c2 cardiomyocytes, we examined OPN’s interaction with the G_α_s/olf protein subunit, previously reported to occur in osteoblastic cells resulting in β_2_AR dysfunction [29]. Indeed, co-immunoprecipitation experiments indicated that OPN and the G_α_s subunit protein interact with each other constitutively (in the absence of any stimulus)—albeit weakly—in H9c2 cardiac cells (Figure 5A). Furthermore, this interaction was unaffected by β_2_AR activation, since salbutamol neither enhanced nor reduced it (Figure 5A,B). Thus, results indicate that similarly to bone cells, OPN physically interacts with G_α_s subunits inside cardiac cells, impeding downstream β_2_AR signaling to AC, cAMP, and Epac1.

## 3. Discussion

In the present study, to the best of our knowledge we report for the first time the adverse role OPN plays in hindering β_2_AR-dependent, cAMP/Epac1-mediated anti-fibrotic signaling and function in rat cardiac myocyte-like cells. By physically interacting constitutively with the G_α_ subunit of the Gs heterotrimeric protein, intracellular OPN reduced the cAMP generating capacity of the β_2_AR in cardiac cells, thereby reducing both the activity (acutely) and protein expression (long-term) of the cAMP effector Epac1, which is normally anti-fibrotic in several tissues, including the heart (Figure 6) [16]. Additionally, we found that aldosterone—a powerful pro-fibrotic and adverse remodeling-associated hormone in the heart that is also elevated during post-MI HF progression [30]—directly induced OPN transcriptional upregulation to mediate fibrosis (Figure 6). Notably, this aldosterone-induced OPN upregulation was blocked by the β_2_AR, so it appears that there was a closed negative feedback loop operating in cardiac myocytes, in which OPN was on one hand induced by aldosterone to promote fibrosis, in part via blockade of β_2_AR-dependent, cAMP/Epac1-mediated signaling, and on the other hand, the β_2_AR opposed the aldosterone/MR-mediated OPN induction (Figure 6).

OPN is a known downstream transcriptional target of aldosterone and its receptor, the steroid-responsive transcription factor MR [20,28]. Immediate/early OPN mRNA induction promoting fibrosis in response to aldosterone has been reported in the kidneys [20], during VSMC proliferation and inflammation [31], in vascular endothelial cells [32], and in peripheral blood mononuclear cells [33]. Moreover, OPN deletion has been shown to ameliorate cardiac pathology post-pressure overload [34] and kidney function in renal failure [35], and a very recent study indicated that OPN may serve as a biomarker for stage stratification of the functional myocardial contractile deficit in chronic angina patients [36]. Other studies have implicated OPN in aortic valve sclerosis/stenosis [37] and in neoangiogenesis [38]. Here we report that OPN was also involved in aldosterone-dependent fibrosis in cardiac myoblasts.

OPN is a ubiquitous, pro-inflammatory cytokine that is activated in response to a variety of hemodynamic, pro-inflammatory, oxidative-stress-related, and pro-fibrotic (e.g., TGFβ) stimuli. It is found both intracellularly and extracellularly, and thus it can promote fibrosis via both intracellular TGFβ signaling facilitation and modulation of extracellular matrix remodeling [29]. Our present findings add a novel mechanism for OPN’s pro-fibrotic effects, at least in the heart— interference with the anti-fibrotic actions of the catecholamines through the β_2_AR. In contrast to the β_1_AR, the β_2_AR is considered cardio-protective in the post-MI failing heart, since it can facilitate infarct (wound) healing; promote cardiomyocyte survival; and limit inflammation, apoptosis, and other adverse remodeling processes that ensue immediately after an MI [4,39,40]. Indeed, cAMP and its effector Epac1—which are induced by the activated β_2_AR in the heart—are known to exert anti-fibrotic effects in several cell types and tissues, including in the heart [16]. Our present findings indicate that OPN reduces β_2_AR-dependent cAMP generation—and thus Epac1 levels and activity—in heart cells by physically interacting with the G_α_s protein (i.e., the cognate signal transducer to which the β_2_AR couples), in order to activate AC and induce cAMP synthesis (Figure 6). This is in line with a previous study in bone cells reporting the exact same mechanism (β_2_AR-dependent cAMP generation hindrance) underlying OPN’s role in the modulation of the sympathetic tone of bone mass regulation [29].

Of course, how exactly OPN inhibits β_2_AR signaling to cAMP in cardiac myocytes remains to be elucidated in future studies. One plausible mechanism could be the recruitment of GPCR-kinase (GRK)-2 (or some other GRK) to the OPN–G_α_s complex. GRKs bind agonist-activated GPCRs and phosphorylate them to induce their functional desensitization (i.e., decoupling from G proteins) [41]. In fact, GRK2, the major GRK isoform regulating the cardiac β_2_AR [4], has been reported to directly phosphorylate and inhibit Epac1 in the central nervous system of mice in vivo, thereby mitigating persistent and chronic inflammatory pain promoted by Epac1-to-Rap1 signaling [42]. Thus, GRK2 could have a dual role in the inhibition of cardiac β_2_AR’s anti-fibrotic signaling: (a) direct desensitization (Gs protein decoupling) of the β_2_AR itself, and (b) direct blockade of Epac1’s anti-fibrotic activity. Identification of the complete mechanism and of the additional molecular partners through which OPN opposes β_2_AR anti-fibrotic signaling to cAMP/Epac1 is definitely worth pursuing, and is the goal of our currently ongoing studies.

Another major question arising from our present findings pertains to the mechanism by which the β_2_AR, in a negative feedback regulatory manner, opposes the aldosterone/MR-dependent OPN upregulation in cardiac myocytes (Figure 6). How does the β_2_AR (a GPCR) block this aldosterone/MR transcriptional effect? The MR is a ~1000-amino acid cytoplasmic (at rest) protein with three functional domains: the N-terminal domain (NTD) that regulates transcriptional activity of the receptor; the DNA-binding domain (DBD) involved in the binding of the promoter of the target gene; and the ligand-binding domain (LBD) responsible for hormone binding [43,44]. In the nucleus, the MR depends on numerous molecular co-regulators to activate and regulate its target genes that carry the (shared with the glucocorticoid receptor) glucocorticoid response element (GRE) sequence in their promoters [45]. Importantly, the MR undergoes various post-translational modifications, such as phosphorylation, ubiquitination, etc., which play important roles in the regulation of its nuclear translocation and of its transcriptional activity [46]. Indeed, the MR contains several serine and threonine residues that are substrates for kinases like PKA, whose phosphorylation of the MR has been reported to inhibit MR cytoplasm-to-nucleus trafficking [47,48]. Given that PKA is activated by the β_2_AR (Figure 6), β_2_AR may inhibit MR-dependent OPN upregulation through PKA. Another possibility is that GRK5, the other major GRK isoform in cardiac cells [4,41], which is also activated by the agonist-occupied β_2_AR [44], inhibits the MR by directly phosphorylating it (A. Lymperopoulos, unpublished data and [27]). We plan to investigate these possibilities and to delineate the exact mechanism of β_2_AR’s regulation of aldosterone signaling in the heart in future studies.

In summary, our results suggest that the ubiquitous pro-inflammatory cytokine OPN, transcriptionally induced by aldosterone and the MR, mediates cardiac fibrosis in part via perturbation of the β_2_AR’s cAMP/Epac1-dependent anti-fibrotic signaling in rat cardiac myoblasts. The β_2_AR, in turn, may inhibit the aldosterone-induced OPN upregulation in the heart, thereby closing a feedback regulation loop. These findings provide a previously unappreciated mechanism for OPN’s cardiac effects and suggest a direct involvement of OPN in cardiac βAR function and signaling. Finally, from a translational/therapeutic standpoint, our data suggest that OPN blockade, perhaps with a cell-permeable anti-OPN antibody or with OPN siRNA-mediated knockdown, can significantly boost the therapeutic efficacy of β_2_AR agonists, already available in clinical practice, or of cAMP analogs and/or Epac1 small-molecule activators (currently in therapeutic development) against cardiac fibrosis and adverse remodeling in general.

## 4. Materials and Methods

### 4.1. Materials

All chemicals (aldosterone, isoproterenol, salbutamol, forskolin) were from Sigma-Aldrich (St. Louis, MO, USA), except for human recombinant TGFβ_1_, which was purchased from Cell Signaling Technology (Danvers, MA, USA).

### 4.2. Cell Culture and CRISPR-Mediated OPN KO

The H9c2 rat cardiomyoblast cell line was purchased from American Type Culture Collection (Manassas, VA, USA) and cultured as previously described [49]. For CRISPR/Cas9-mediated OPN genetic deletion, a custom-made oligo targeting an mRNA exon of the rat OPN gene (Target mRNA RefSeqId: NM_012881/6, *Spp1* gene of the Rattus norvegicus species) was designed, synthesized, and inserted into a CRISPR lentiviral construct (Sigma-Aldrich). Forty-eight hours after infection of H9c2 cells with this rat OPN-specific CRISPR lentivirus, protein extracts were prepared and the knockdown/knockout of OPN was verified via Western blotting for OPN with an anti-rat OPN monoclonal antibody (MPIIIB10(1); Developmental Studies Hybridoma Bank-DSHB, Iowa City, IA, USA).

### 4.3. Real-Time qPCR

Real-time quantitative PCR for OPN, collagen type I, III, and IV, and fibronectin was performed essentially as described [50,51]. Briefly, total RNA was isolated from H9c2 cells with the Trizol reagent, according to the manufacturer’s instructions (Invitrogen, Carlsbad, CA, USA), followed by quantitative real-time PCR in a MyIQ Single-Color Real-Time PCR detection system (Bio-Rad Life Sciences Research, Hercules, CA, USA) using SYBR Green Supermix (Bio-Rad) and 100 nM of gene-specific oligonucleotides. Quantification of mRNA included normalization to 18s rRNA levels. No bands were seen in control reactions in the absence of reverse transcriptase. Primer pairs used were (all from Sigma-Aldrich) [20]: 5′-TGGCAGTGGTTTGCTTTTGC-3′ and 5′-CCAAGTGGCTACAGCATCTGA- 3′ for OPN; 5′-ATCTCCTGGTGCTGATGGAC-3′ and 5′-ACCTTGTTTGCCAGGTTCAC-3′ for collagen type I; 5′-AGGCAACAGTGGTTCTCCTG-3′ and 5′-GACCTCGTGCTCCAGTTAGC-3′ for collagen type III; 5′-GGCGGTACACAGTCAGACCAT-3′ and 5′-TGGTGTGCATCACGAAGGA-3′ for collagen type IV; 5′-CGAGGTGACAGAGACCACAA-3′ and 5′-CTGGAGTCAAGCCAGACACA-3′ for fibronectin; and 5′-TCGATGCTCTTAGCTGAGTG-3′ and 5′-TGATCGTCTTCGAACCTCC-3′ for 18S rRNA.

### 4.4. cAMP Accumulation Determination

cAMP accumulation was measured with the “Direct cAMP ELISA kit” (Product #ADI-900-066; Enzo Life Sciences, Farmingdale, NY, USA), essentially as described previously [50].

### 4.5. Co-Immunoprecipitation and Western Blotting

H9c2 cell extracts were prepared, as described previously [49], in a 20 mM Tris pH 7.4 buffer containing 137 mM NaCl, 1% Nonidet P-40, 20% glycerol, 10 mM PMSF, 1 mM Na_3_VO_4_, 10 mM NaF, 2.5 µg/mL aprotinin, and 2.5 µg/mL leupeptin. Protein concentration was determined and equal amounts of protein per sample were used for immunoprecipitation (IP) or Western blotting. Epac1 was detected via Western blotting in total cellular extracts with an anti-Epac1 antibody (sc-28366; Santa Cruz Biotechnology, Santa Cruz, CA, USA), coupled with immunoblotting for GAPDH with an anti-GAPDH antibody (sc-25778; Santa Cruz Biotechnology) as protein loading control. For the OPN–G_α_s co-IPs, OPN was immunoprecipitated by overnight incubation of H9c2 cell protein extracts with the mouse anti-OPN antibody (DSHB) attached to Protein A/G-Sepharose beads (Sigma-Aldrich). The IPs were then subjected to immunoblotting for G_α_s/olf protein (sc-55545; Santa Cruz Biotechnology), and for OPN again, to confirm IP of equal amounts of endogenous OPN. All immunoblots were revealed by enhanced chemiluminescence (ECL, Life Technologies, Grand Island, NY, USA) and visualized in the FluorChem E Digital Darkroom (Protein Simple, San Jose, CA, USA), as described previously [6,49,50,51,52]. Densitometry was performed with the AlphaView software (Protein Simple) in the linear range of signal detection (on non-saturated bands).

### 4.6. Statistical Analysis

Data are generally expressed as means ± S.E.M. Unpaired two-tailed Student’s *t*-test and one-way ANOVA with Bonferroni test were performed for statistical comparisons. For all tests, a *p*-value < 0.05 was generally considered to be significant.

## Figures and Tables

**Figure 1 ijms-20-01396-f001:**
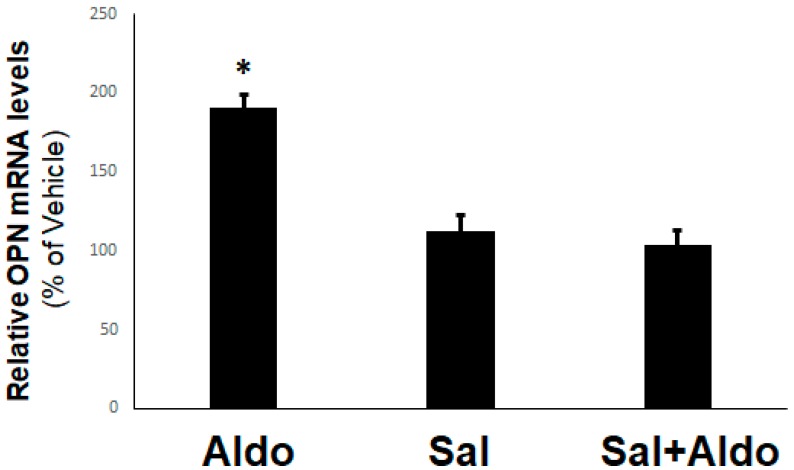
β_2_-adrenergic receptor (AR) inhibits aldosterone-induced osteopontin (OPN) upregulation in H9c2 cardiomyocytes. H9c2 cells were treated with 10 nM aldosterone (Aldo), 10 µM salbutamol (Sal), or 10 nM aldosterone in the presence of 10 µM salbutamol (Sal+Aldo) for 2 h. At the end of this 2-h period, cells were harvested, total RNA isolated, and real-time PCR for OPN mRNA quantitation was performed. *, *p* < 0.05, vs. any other treatment; n = 4 independent experiments/condition.

**Figure 2 ijms-20-01396-f002:**
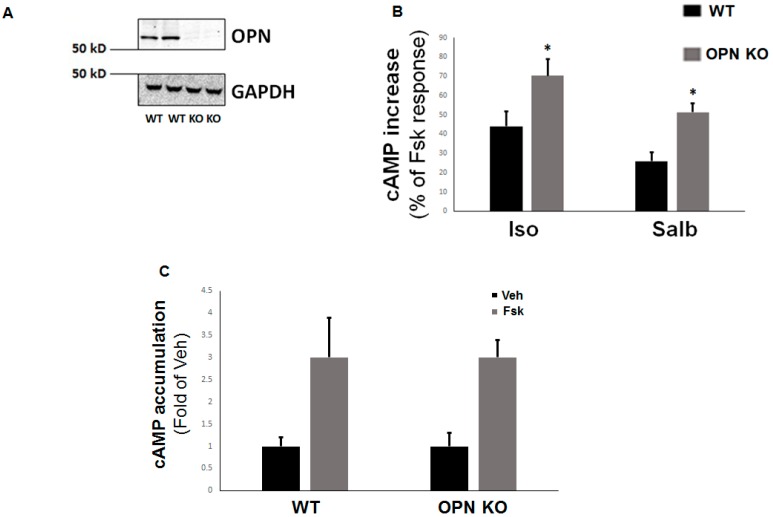
Enhanced β_2_AR-dependent cAMP accumulation in the absence of OPN in H9c2 cardiomyocytes. (**A**) Western blotting to confirm OPN CRISPR-mediated deletion in H9c2 cells infected with OPN-specific CRISPR lentivirus (knockout, KO) or mock CRISPR lentivirus (wild type, WT). Blotting for glyceraldehyde 3-phosphate dehydrogenase (GAPDH) is also shown as loading control. Representative blots from three independent experiments per condition with similar results are shown. (**B**) cAMP accumulation in response to 10 µM isoproterenol (Iso) or 10 µM salbutamol (Salb) in control WT and in OPN-depleted (OPN KO) H9c2 cells, expressed as percent of the respective cAMP production induced by 10 µM forskolin (Fsk). *, *p* < 0.05, vs. WT; n = 3 independent experiments/condition/cell clone. (**C**) cAMP accumulation in response to 10 µM forskolin (Fsk) or vehicle (Veh) in control WT and in OPN-depleted (OPN KO) H9c2 cells. No significant differences were observed between WT-Fsk and OPN KO-Fsk at *p* = 0.05 (n = 3 independent experiments/condition/cell clone).

**Figure 3 ijms-20-01396-f003:**
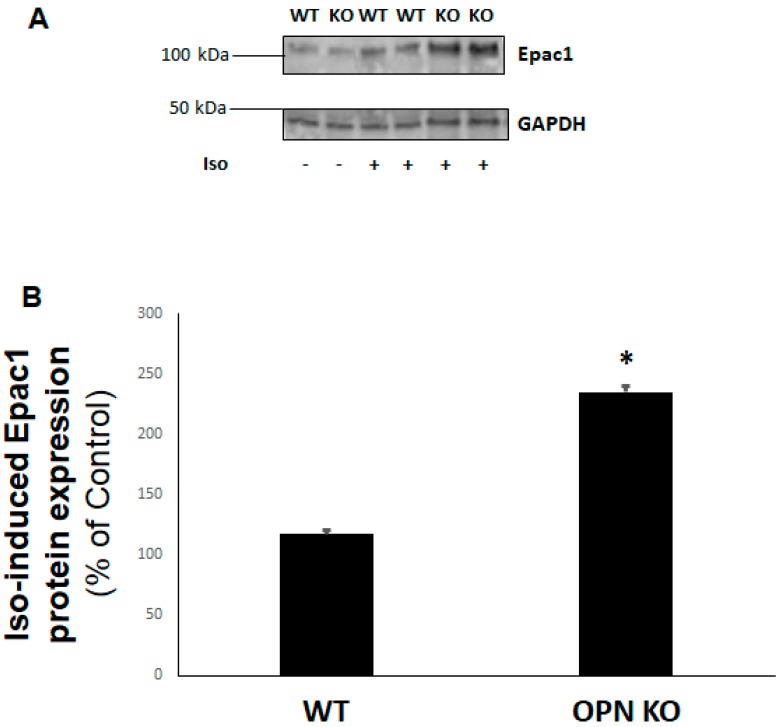
Enhanced βAR-dependent Epac1 protein levels in the absence of OPN in H9c2 cardiomyocytes (Epac: exchange protein directly activated by cAMP). H9c2 cells were treated with 10 µM isoproterenol (Iso) or vehicle for 24 h in the presence (WT) or absence (KO) of OPN, then cells were harvested and total protein extracts prepared for Epac1 immunoblotting. Representative blots are shown in (**A**), including GAPDH as loading control, and the densitometric quantitation of three independent experiments per condition performed in duplicate is shown in (**B**). *, *p* < 0.05, n = 3.

**Figure 4 ijms-20-01396-f004:**
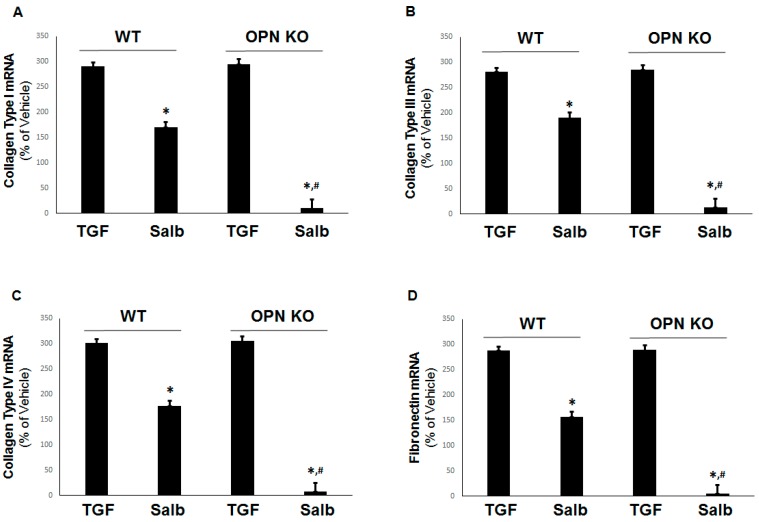
Absence of OPN potentiates the β_2_AR-mediated inhibition of TGFβ-dependent profibrotic factor mRNA induction in H9c2 cells. mRNA levels of (**A**) type I collagen, (**B**) type III collagen, (**C**) type IV collagen, and (**D**) fibronectin, in WT or OPN-depleted (OPN KO) H9c2 cells treated with 10 ng/mL TGF-β_1_ (TGF) with or without 10 µM salbutamol (Salb). *, *p* < 0.05, vs. TGF; ^#^, *p* < 0.05, vs. WT-Salb; n = 3 independent experiments per condition (two-way ANOVA with Bonferroni test).

**Figure 5 ijms-20-01396-f005:**
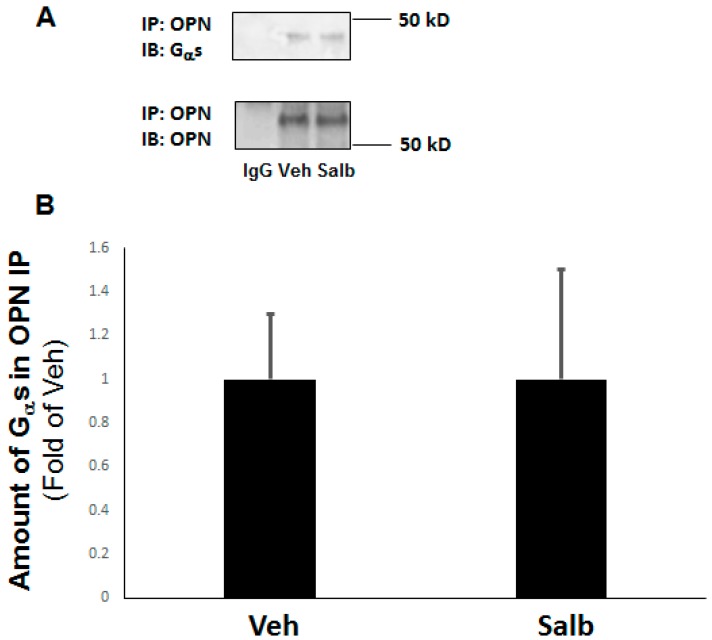
OPN opposes β_2_AR signaling via physical interaction with G_α_s in H9c2 cardiomyoblasts. Co-immunoprecipitation (co-IP) of OPN with G_α_s in native WT H9c2 cells treated with vehicle (Veh) or 10 µM salbutamol (Salb). Representative blots are shown in (**A**), and the densitometric quantitation of three independent experiments is shown in (**B**). IB: immunoblotting; IP: immunoprecipitation; IgG: IP with a general IgG antibody (negative control for the OPN IP). No significant difference (at *p* = 0.05) in the amount of G_α_s co-IP’d with OPN was observed between Veh and Salb (n = 3).

**Figure 6 ijms-20-01396-f006:**
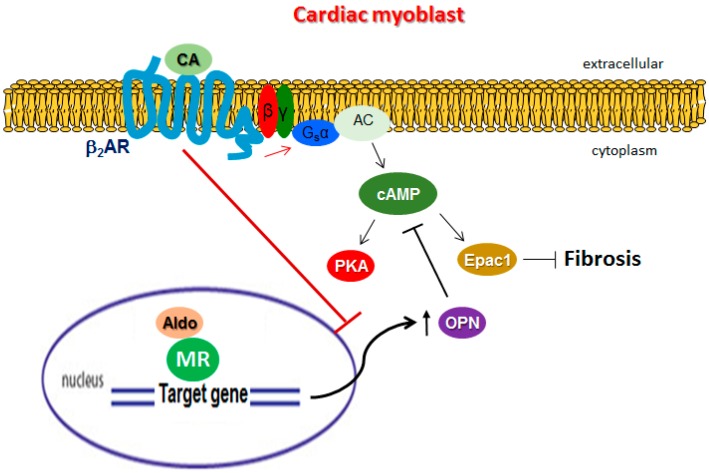
Schematic illustration of the proposed role of OPN in β_2_AR ant-fibrotic signaling in H9c2 cardiomyocytes. CA: catecholamine; MR: mineralocorticoid receptor. See text for details and for all other molecular acronym descriptions.

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
