# Peer review of "Deletion of Osteopontin Enhances β2-Adrenergic Receptor-Dependent Anti-Fibrotic Signaling in Cardiomyocytes"

_ijms, 2019, doi:10.3390/ijms20061396_

Reviewer 1 Report

The manuscript is very interesting and well written. I just have some editorial comments:

The potential utilization of osteopontin as biomarker of ischemic heart disease (Osteopontin - a biomarker of disease, but also of stage stratification of the functional myocardial contractile deficit by chronic ischemic heart disease. J Enzyme Inhib Med Chem. 2019 Dec;34(1):783-788. doi: 10.1080/14756366.2019.1587418) should be discussed.

The mechanistic role of osteopontin in the regulation of endothelial cell function (J Cell Physiol. 2011 Aug; 226(8): 2139–2149; Oncogene. 2009 Sep 24;28(38):3412-22) should be discussed

Line 55: when mentioning adrenergic receptors, an updated overview    DOI: 10.1016/B978-0-12-803111-7.00011-7 should be quoted.

The Authors should comment on the clinical significance of their discovery, especially in terms of the drugs currently available for the modulation of beta adrenergic receptors (Table 5.1 of DOI: 10.1007/978-3-319-15961-4_5).

Author Response

We thank this reviewer for his/her invaluable comments that have helped us improve our manuscript significantly. We have added the discussions of the pertinent citations the reviewer suggested in the revised version of our manuscript (all changes highlighted in yellow in the manuscript file). We hope this satisfies this reviewer.

We wish to thank him/her again for their comments and suggestions.

Reviewer 2 Report

Fibrosis is a serious pathologic problem in the body. The manuscript aims to investigate the role of pro-inflammatory cytokine osteopontin (OPN) in beta2-adrenergic receptor (AR)-regulated cardiomyoblast fibrosis using H9c2 rat cardiomyoblast cell line. 

It is currently interesting subject and all data are reliable. However, this reviewer has some specific idea and questions below.

1. In this study, the authors uncovered a direct inhibitory role of OPN in cardiac beta2-AR anti-fibrotic signaling via cAMP/Epac1. Their data suggest that OPN blockade, perhaps with a cell-permeable anti-OPN antibody or with OPN siRNA-mediated knockdown, can significantly boost the therapeutic efficacy of beta2-AR agonists, or of cAMP analogs and/or Epac1 small molecule activators against cardiac fibrosis and adverse remodeling in general. Thus, the authors have just demonstrated  the role of OPN but not the methods of OPN blockade even the authors used the CRISPR-mediated OPN deletion in H9c2 cells. For this reviewer, the title "Deletion of osteopontin enhances..." is not suitable for this manuscript. Please reconsider it along the author's results and future directions.

2. In Figure 4, TGFbeta treatment increased mRNA expressions of collagen type I, type III, type IV, and fibronectin in both WT and OPN KO cells. The authors showed that Salb treatment decreased the TGFbeta-increased mRNA expressions of all of these pro-fibrotic factors. Especially in WT cells, however, this reviewer wonders whether mRNA expression of OPN is decreased or not in TGFbeta+Salb-treated WT cells. This reviewer has no idea whether mRNA expression of OPN is increased or not in TGFbeta-treated WT cells. Please clarify these questions. 

3. In Figure 4, the authors showed the results of statistical analyses that are compared with WT-Salb. This reviewer suggest the importance of comparisons between WT-Salb and OPN KO-Salb to demonstrate the role of OPN in TGFbeta-increased mRNA expressions of pro-fibrotic factors. Please mention the author's idea. 

Author Response

We thank this reviewer for his/her invaluable comments that have helped us to significantly improve our manuscript. Below is our point-by-point response:

1) The reviewer is correct about the comments on OPN blockade. Since we did not actually block OPN pharmacologically, but merely deleted its gene via CRISPR, we opted for the current title talking about "deletion of OPN" rather than "Blockade of OPN". We feel this is the most accurate title to use to describe our present findings and thus, the most appropriate one. We hope the reviewer agrees.

2) This is a very valid point raised by the reviewer and we thank him/her for that. We cannot exclude the possibility (which is, in fact, quite likely) that TGFbeta upregulates OPN itself in WT cells. However, our goal in these experiments was to measure mRNA levels of the collagen types and of fibronectin, not of OPN itself. Also, precisely to account for any confounding of the qPCR results by effects of TGFbeta on OPN itself, we analyzed and present our data as % of the vehicle response instead of % of the TGFbeta responses. Therefore, whatever TGFbeta does on OPN levels in WT cells, the effect should be the same between TGFbeta and TGFbeta+Salb WT cells. We have also added a sentence clarifying this very pertinent point raised by this reviewer in the text of our revised manuscript, as follows (lines 199-203 of the revised text, also highlighted in yellow):

"Of note, the partial inhibition of TGFb-induced pro-fibrotic mediators by the b2AR in WT cells might be due, in part, to TGFb-induced transcriptional upregulation of OPN itself, which could raise the “baseline” mRNA levels of collagen and fibronectin the b2AR would need to suppress in these cells."

We hope this now satisfies this reviewer.

3) This is another very valid point raised by the reviewer and we thank him/her for this. We have compared WT-Salb vs. OPN KO-Salb in Fig. 4, as the reviewer suggests. Again, our goal in those experiments was to explore the role of OPN in the salbutamol`s inhibition of fibrosis, not in TGFbeta`s induction of fibrosis per se. In any case, the reviewer is referred to our response to comment #2 above. We believe the sentence we have now added in the text of the revised manuscript about this figure (Figure 4) addresses both comments #2 & #3 raised by this reviewer.

Once again, we wish to thank this reviewer for his/her very constructive comments that we believe have substantially improved the quality of our work.

Round  2

Reviewer 1 Report

No further comments

Reviewer 2 Report

The authors have reasonably replied to my requests.